# Carbon Nanotube Composites with Bimetallic Transition Metal Selenides as Efficient Electrocatalysts for Oxygen Evolution Reaction

Shamas Riaz [1], Muhammad Shafiq Anjum [2], Abid Ali [3,*], Yasir Mehmood [4], Muhammad Ahmad [1], Norah Alwadai [5,*], Munawar Iqbal [1], Salih Akyürekli [6], Noor Hassan [7] and Rizwan Shoukat [8,*]

1. Department of Chemistry, Division of Science and Technology, University of Education, Lahore 54770, Pakistan; shamasriaz33221@gmail.com (S.R.); dr.muhammad.ahmad@ue.edu.pk (M.A.); munawar.iqbal@chem.uol.edu.pk (M.I.)
2. Department of Physics, Government College University, Lahore 54000, Pakistan; shafiqanjum55@gmail.com
3. Department of Chemistry, The University of Lahore, 1-Km Defence Road, Lahore 54590, Pakistan
4. Department of Chemistry, Abbottabad University of Science and Technology, Abbottabad 22500, Pakistan; yasirmehmood.qau@gmail.com
5. Department of Physics, College of Sciences, Princess Nourah bint Abdulrahman University, P.O. Box 84428, Riyadh 11671, Saudi Arabia
6. Department of Physics, Faculty of Arts and Sciences, Süleyman Demirel University, 32260 Isparta, Turkey; salihakyurekli@gmail.com
7. College of Chemistry and Life Sciences, Zhejiang Normal University, Jinhua 321004, China; raviannoor80@hotmail.com
8. Department of Mechanical, Chemical and Materials Engineering, University of Cagliari, Via Marengo 2, 09123 Cagliari, CA, Italy
* Correspondence: abid.ali@chem.uol.edu.pk (A.A.); norah.alwadai@hotmail.com (N.A.); rizwan.shoukat@unica.it (R.S.); Tel.: +92-3215051352 (A.A.); +39-3474752100 (R.S.)

**Abstract:** Hydrogen fuel is a clean and versatile energy carrier that can be used for various applications, including transportation, power generation, and industrial processes. Electrocatalytic water splitting could be the most beneficial and facile approach for producing hydrogen. In this work, transition metal selenide composites with carbon nanotubes (CNTs) have been investigated for electrocatalytic water splitting. The synthesis process involved the facile one-step hydrothermal growth of transition metal nanoparticles over the CNTs and acted as an efficient electrode toward electrochemical water splitting. Scanning electron microscopy and XRD patterns reveal that nanoparticles were firmly anchored on the CNTs, resulting in the formation of composites. The electrochemical measurements reveal that CNT composite with nickel–cobalt selenides (NiCo-Se/CNTs@NF) display remarkable oxygen evolution reaction (OER) activity in basic media, which is an important part of hydrogen production. It demonstrates the lowest overpotential ($\eta_{10\text{mAcm}^{-2}}$) of 0.560 V vs. RHE, a reduced Tafel slope of 163 mV/dec, and lower charge transfer impedance for the OER process. The multi-metallic selenide composite with CNTs demonstrating unique nanostructure and synergistic effects offers a promising platform for enhancing electrocatalytic OER performance and opens up new avenues for efficient energy conversion and storage applications.

**Keywords:** metal selenides; CNT composites; electrocatalyst; oxygen evolution reactions

## 1. Introduction

Society, the economy, and the environment are the three interconnected dimensions at the core of sustainability, drawing significant attention and focus. Two major issues for modern civilization are energy problems and environmental deprivation [1–3]. Therefore, humanity greatly depends on the rational design of highly active, stable, and cost-effective electrocatalysts for electrochemically converting water into $H_2$ and $O_2$. Electrochemical

water conversion has been recognized as a promising approach for advancing environmentally friendly and green renewable energy sources, functioning as a viable alternative to depleting fossil energy resources [4–6]. In this technology, the cathode produces high-energy-density hydrogen ($H_2$) with zero pollution, making it a clean and renewable energy source with no harmful impact on the environment [7–9]. At the anode, the oxygen evolution reaction (OER) occurs with slow kinetics and a sluggish four-electrode process [10–12]. The OER plays a vital role in the development of new hybrid energy technologies, such as solar-to-fuel energy production and metal–air hydroelectric batteries [13–15]. The electrochemical water-splitting process is notably different from the direct conversion of solar energy to hydrogen and remains a significant challenge [16]. Undoubtedly, OER represents a significant hindrance to enhancing the overall conversion efficiency of water electrolysis. This is due to the complex four-electron and four-proton transfer process, which necessitates an extremely high overpotential to surmount the kinetic barrier of OER [17–19]. Consequently, the utilization of highly effective electrocatalysts is essential for water electrolysis to reduce the significant overpotential and accelerate the rate of oxygen evolution by lowering the activation energy barrier. Simultaneously meeting the demand for fabricating and developing catalytically active sites or robust structures is crucial to activating the sluggish oxygen evolution reaction (OER).

In response to these challenges, there has been extensive research focused on designing OER electrocatalysts with robustness, long-term stability, and economic efficiency [20–22]. Among these, esteemed metal oxides, such as iridium dioxide ($IrO_2$) and ruthenium dioxide ($RuO_2$), have demonstrated exceptional catalytic activity and low overpotential for (OER) in basic media [23,24]. Recently, a cost-effective oxygen evolution catalyst was fabricated using earth-abundant transition elements, including Co [25], Fe [26], Zn [27], Mn [28], and Ni [29], which has been notably encouraging. While the OER activity often outperforms that of the corresponding single-metal catalysts and their oxides, most of them suffer from issues related to catalyst stability. Therefore, research efforts have predominantly focused on synthesizing and systematically modulating the 3D–electronic structure of transition metal chalcogenides (3d-TMCs) like sulfides [30], phosphides [31], and selenides [32]. These materials exhibit higher conductivities and improved OER activities compared to their corresponding oxide/(oxy)hydroxide counterparts in alkaline conditions [33]. Specifically, 3d-TMCs, mixed ternary bimetallic selenides, have exhibited superior electro-catalytic performance. This improvement is attributed to the smaller electronegativity of selenium, which results in an enlarged covalent character in the metal–ligand interaction, facilitating catalyst activation [34,35]. Chen et al. explored the Co-Ni-$Se_2$ nanostructure catalyst, which offers numerous active centers, a large surface area, and synergistic effects due to the presence of multiple metal constituents, enhancing its efficiency in water splitting [4]. Subsequently, Zhao X. et al. designed a highly effective multi-metal Co-Mn-$Se_2$ nano-architecture catalyst capable of generating active species and significantly enhancing the OER rate [36]. Hence, transition metal selenides, especially mixtures of 3D transition multi-centered metal (Ni, Mn, Fe, Cu, Co, etc.)-based selenides, have greatly facilitated catalyst activity and ensured exceptional performance in OER by tuning the electronic charge density of neighboring dopant atoms [37,38]. Nevertheless, the main hindrance plaguing these materials in industrial applications is their inadequate electrical conductivity and low intrinsic OER activity in alkaline media [39–41]. Hence, rational efforts are essential to further enhance the specific surface area and electrical conductivity of these materials for their scalable application.

In pursuit of this objective, there has been a persistent interest in designing appropriate 3D nanostructures of multi-metal selenides and their hybridization with a conductive matrix, such as carbon nanotubes (CNTs) grown on nickel foam, to attain improved electrocatalytic activity with unique electronic structure; this allows them to offer high catalytic activity at the active sites, facilitating efficient electron/ion transformation [22,40,42]. Huang et al. presented a one-step phosphorization and selenization method to synthesize a distinctive Co-P-Se ternary hybrid integrated with $CoSe_2$/CNTs (CoPSe-$CoSe_2$/CNTs), which significantly

enhanced the catalytic performance for the oxygen evolution reaction (OER) [43]. H. Sun et al. discussed few noble and non-noble metals for green hydrogen production [44]. Yuan M. et al. developed hydrothermal cobalt selenide tuned with carbon nanotube (CoSe2@C-CNT) for the oxygen evolution reaction (OER) in basic media with higher catalytic activity and better stability [45]. Yang et al. synthesized nickel–cobalt–phosphide (NiCoP) supported on feather-like structures on nickel foam (NF), which was achieved via a two-step hydrothermal phosphorization process for clean energy by exceptional insights into hydrogen production via electrolysis, employing simple and effective preparative methods. The experimental findings and DFT calculations jointly indicate that NiCoP reveals a heightened Density of States (DOS) at the Fermi level, thereby promoting enhanced charge transfer. This effect is observed specifically at 203 mV over-potentials of 44, achieving a current density of 10 mA $cm^{-2}$ in a 1 M KOH for both the hydrogen evolution reaction (HER) and oxygen evolution reaction (OER) [46]. Using a DES for bimetallic sulfide fabrication offers a promising solution to minimize dependence on toxic solvents. This method not only alters the electronic structure but also introduces a nanostructure, leading to the creation of multiple active sites. Moreover, it improves conductivity, aids in gas-evolution behavior, and optimizes adsorption energy with intermediates, ultimately enhancing the performance of OER and HER. The OER performance of $NiCo_2S_4$ was significantly improved, displaying enhanced activity and stability. $NiCo_2S_4$ was synthesized via a one-pot hydrothermal treatment at 433.15 K for 16 h, and the solvent employed for this synthesis was a PEGylated (DES). $NiCo_2S_4$ had higher OER activity and lower charge-transfer resistance than $NiS_2$ and $CoS_2$, and the structural study indicated that $NiCo_2S_4$ is formed of a cubic phase of ions. Ni is consistently distributed inside $NiCo_2S_4$. The $NiCo_2S_4$ indicated its high activity for OER in 1M KOH at 10 mA $cm^{-2}$ less overpotential of 337 mV, and its negligible decay highlights its notable stability and durability at 50 mA $cm^{-2}$ even after 2000 CV [47]. Recently, Zhang et al. prepared FeCoMnNiMOF-74/NF hybrid on nickel foam hydrothermally and unveiled that the synergistic effect of Ni, Mn, and Fe in the composite significantly enhanced the OER activity [48].The exploration of mixed metal selenides phase and their 3D nanocomposites have been limited due to the scarcity of preparation techniques for manipulating the nanoscale structure and morphology of multi-metal-based selenides to achieve a tailored nano-architecture.

In this study, we present a facile one-step hydrothermal method to synthesize composites of CNTs and multi-metal selenides. Taking advantage of their high conductivity, large surface area, and enhanced active centers, multi-metal selenide composites could potentially serve as fascinating electrocatalysts. The electrochemical measurements reveal that the multi-metallic selenides and their composites (NiCo-Se/CNTs@NF) exhibited superior OER activity in alkaline electrolytes, characterized by the lowest overpotential (OER) of 0.560 V (vs. RHE), a Tafel slope of 163 mVdec$^{-1}$, and reduced charge transfer resistant compared to other binary metal selenides. The 3D multilevel hexagonal-type nanostructure fulfills the requirements for both remarkable electrocatalytic activity coupled with exceptional long-term stability. In a long-term stability test, the fabricated composite electrode (NiCoSe/CNTs@NF) maintained a constant current density of ~10 mA/cm$^2$ ($\eta_{10}$) in a 1 M KOH solution for approximately 15 h, demonstrating excellent electrochemical stability. Therefore, the remarkable electrocatalytic performance signifies that the structure of NiCoSe/CNTs@NF offers a promising strategy to enhance the catalytic OER performance of self-supported electrodes and has potential for various energy conversion applications.

## 2. Experimental

### 2.1. Reagents and Materials

All materials were purchased from Merk. Nickel chloride hexahydrate ($NiCl_2 \cdot 6H_2O$ 99.9%), cobalt chloride hexahydrate ($CoCl_2 \cdot 6H_2O$ 99.9%), selenium powder (Se 99.9%), and CNTs (98%) were used during synthesis. Analytical-grade hydrochloric acid and sulfuric acid were used for the oxidation of CNTs, and deionized water was utilized throughout the experiments.

## 2.2. Synthesis of NiCoSe@CNTs and MnCoSe@CNTs over the Ni-Foam

The oxidation and enhancement of hydrophilicity in carbon nanotubes (CNTs) were conducted via the previously reported method described by Zhongbin et al. [49]. Ni foam was separated into equal parts ($1 \times 1$ cm$^2$) and subsequently ultrasonically pre-treated to remove the impurities. A solution containing acetone, hydrochloric acid (0.1 M), and ethanol was used to remove the surface oxide layer. In brief, 15 mg of CNT was dispersed in 20 mL of deionized water with ultrasonic treatment for 30 min. Following this, a solution containing 4.0 mmol (948 mg) of NiCl$_2$.6H$_2$O and CoCl$_2$.6H$_2$O (952 mg) was added, and the resulting mixture, containing 6 mL of an aqueous solution of selenic acid (0.08 M) with a Co/Se molar ratio of 1:2, was manually transferred into a 50 mL Teflon-lined autoclave for hydrothermal treatment. Activated Ni foam was carefully placed into the autoclave. Subsequently, the autoclave was subjected to a thermal treatment, maintaining a temperature of 160 °C for 24 h. After completion, the autoclave was allowed to cool gradually to reach the ambient temperature. Finally, Ni foam modified with NiCoSe@CNTs (MnCoSe@CNTs) was extracted, subjected to five rounds of washing with ethanol and N-methyl-2-pyrrolidinone at 50 °C, and subsequently dried. A MnCoSe/CNT on nickel foam (MnCoSe/CNTs@Ni) composite was synthesized following a procedure similar to that of NiCoSe/CNTs@Ni, with the exception that nickel chloride hexahydrate was replaced by manganese chloride tetrahydrate (MnCl$_2$·4H$_2$O) at 23.75 mg (0.12 mmol).

## 2.3. Material Characterization

The structure and morphology of catalysts were observed via Quanta FEG 250 scanning electron microscopy (SEM) from FEI company made in Czech Republic. The scanning electron microscope system, equipped with analytical systems, utilized an energy-dispersive spectrometer for EDX analysis. The Quanta 600F scanning electron microscope offers a resolution of 2.5 nm for backscattered images and 1.2 nm for secondary electron images when operating at 10.0 KeV. The structure of the as-prepared material was characterized using a Panalytical X'pert Pro diffractometer (Spectris Company) with Cu K$\alpha$ as the X-ray source at 45 kV and 40 mA for XRD analysis. The 2θ range was scanned from 20° to 80° with a scan rate of 0.01 °/s.

## 2.4. Electrochemical Assessment

All electrochemical analyses for the oxygen evolution reaction (OER) were conducted at room temperature using Gamry Instruments-Reference 3000 potentiostat/galvanostat along with Echem software (analyst). A conventional three-electrode system with 1.0 M KOH as the electrolyte (pH = 13.7) was employed for the experiments. The cleaned NFs coated with CNTs, NiCoSe/CNTs, and MnCoSe/CNTs were utilized as the working electrodes, while a platinum wire and Ag/AgCl (saturated KCl) served as the auxiliary electrode and reference electrode, respectively. All electrocatalytic activity evaluations for the oxygen evolution reaction (OER) were performed after continuous purging of N$_2$ via the 1.0 M KOH electrolyte for 30 min. The potential was calculated with respect to the reversible hydrogen electrode (RHE) by conducting a calibration experiment.

The catalytic activity was evaluated using linear sweep voltammetry (LSV) at a slow scan rate of 5 mV/s in 1 M KOH solution, following ten cycles of cyclic voltammetry (CV) to stabilize the current densities. The measured potential for OER vs. Ag/AgCl was converted to the reversible hydrogen electrode using the following equation: $E_{RHE} = (E_{Ag/AgCl} + 0.198 + 0.0592 \times pH)V$, (pH ~13.6 for all values). The electrochemical active surface areas (ECSAs) of the catalyst were determined from the electrochemical double-layer capacitance ($C_{dl}$) via cyclic voltammetry (CV) in the non-Faradaic potential range at various scan rates, ranging from 50 to 600 mV/s, in the potential range from 0.1 V to 0.20 V vs. Ag/AgCl. The charge transfer resistances ($R_{ct}$) and reaction kinetics at the catalysts/electrolyte interfaces were assessed using electrochemical impedance spectroscopy (EIS) recorded at the open circuit potential (5 mV) in the frequency range of 0.1Hz to 100 MHz. Finally, chronoamperometric measurements were conducted at an overpotential of 0.406 V vs. RHE to maintain a current

density of approximately 10 mA/cm$^2$ for 15 h, evaluating the long-term stability of the electrode material.

## 3. Results and Discussions

### 3.1. Structure, Morphology, and Composition of the As-Prepared Catalyst

X-ray diffraction (XRD) analysis is a powerful technique used to determine the crystallographic structure of materials. Nanomaterials often exhibit unique properties, including being smaller in size and having a high surface area with generated quantum effect realized XRD as particular techniques to examine the structural characteristics. The x-axis represents the diffraction angle (2θ) in degrees, and the y-axis represents the intensity of the diffracted X-rays. The powder X-ray diffraction (XRD) helped to characterize the sample regarding the crystal structure by virtue of radiation created via Cu metal with Kα (λ = 1.54056 Å). The crystallinity and phase structure of the synthesized nanocomposites over the NF provided the diffraction pattern, as shown in Figure 1. The main diffraction peaks in the XRD pattern of CNTs appear at specific 2θ angles with a broad peak at 28.5° and a small hump in the range of 41.8°. These peaks correspond to the (002) and (100) crystallographic planes of CNTs, while the manganese-based CNT composite showed sharp peaks at 35.2°, 38.7°, 50.7°, 57.0°, and 61.8° 70.7°, 72.0°, and 76.6°, which correspond to (311), (400) (311) (311) (311) (311), which is almost equivalent to standard card (JCPDS No. 11-683). In the XRD patterns of the NiCoSe sample, 31.2°, 36.7°, 44.7°, 59.0°, and 65.1°, are compatible with standard (JCPDS No. 08-4821) [50] with MnSe. In addition, no extra diffraction peak was detected, manifesting the high purity of as-prepared samples.

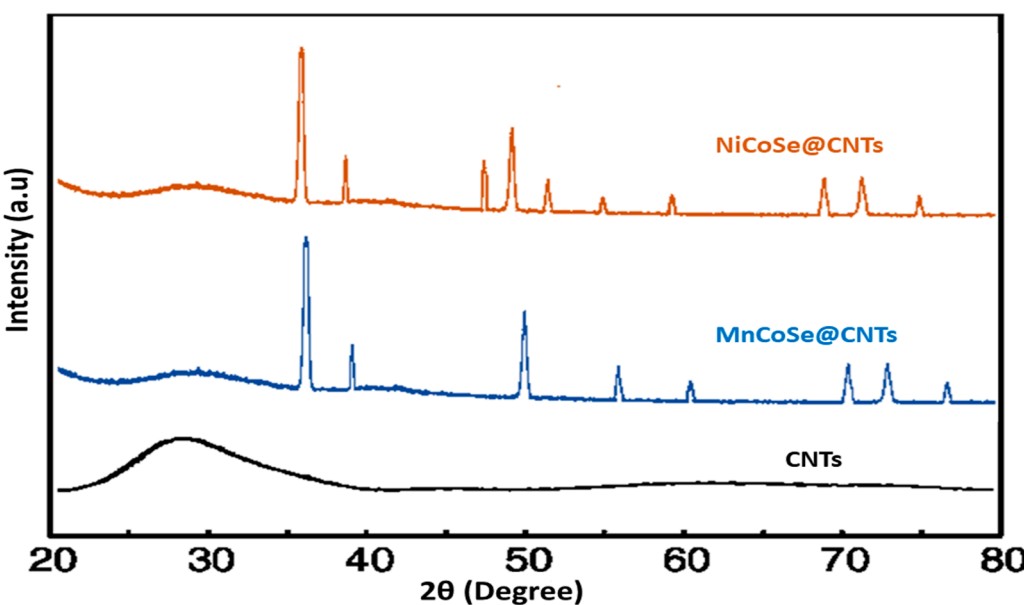

**Figure 1.** X-ray diffraction pattern for the CNTs and their metal selenide composites of MnCoSe@CNTs and NiCoSe@CNTs.

SEM and EDX Analysis

Scanning electron microscopy (SEM) used to observe the structure of materials at micro/nano dimensions. SEM is also able to provide high-resolution images of the surface morphology and structural details of materials at the nanoscale. The key aspects of SEM analysis for nanomaterials could provide high-resolution images, surface roughness, size distribution, and crystallographic information. Energy-dispersive X-ray spectroscopy (EDX) is frequently coupled with SEM to ascertain the elemental mapping and relative abundance of the different elements within the sample. To validate the presence and metal-based selenide distribution of each element over CNTs, SEM images and elemental mappings of the multimetallic NiCoSe@CNTs composites are showcased in Figures 2 and 3. Figure 2a–d

demonstrate the multi-metal selenide nanoparticles (NiCoSe) are uniformly distributed and constitute an interconnected framework of CNTs. In situ growth of NiCoSe@CNTs composites over the NF demonstrated in Figure 2a with the porous structure of NF, while in-depth analysis with higher magnification displayed the modified CNTs network with nanoparticles (NiCoSe). As-synthesized nanocomposites over the porous structure of NF provided a higher surface area, which is extended by the additional surface that comes from the CNTs network to anchor the nanoparticles. The NiCoSe@CNTs nanoparticles formulated excellent active sites with CNTs and are further encapsulated by them, which facilitated the fast electron transfer during the electrochemical process of OER.

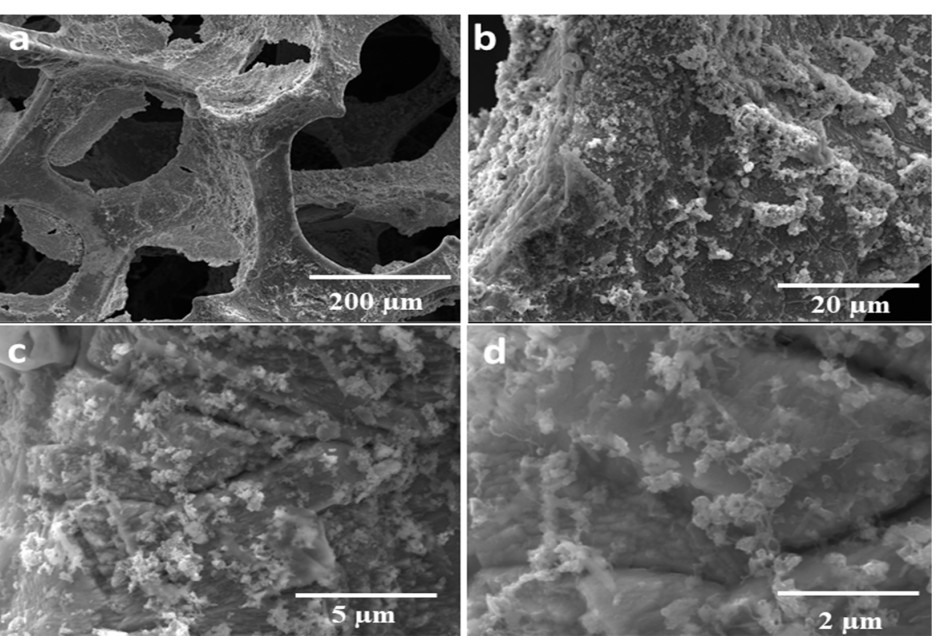

**Figure 2.** SEM mages for NiCoSe@CNTs modified NF with (**a**,**b**) lower and (**c**,**d**) higher magnifications.

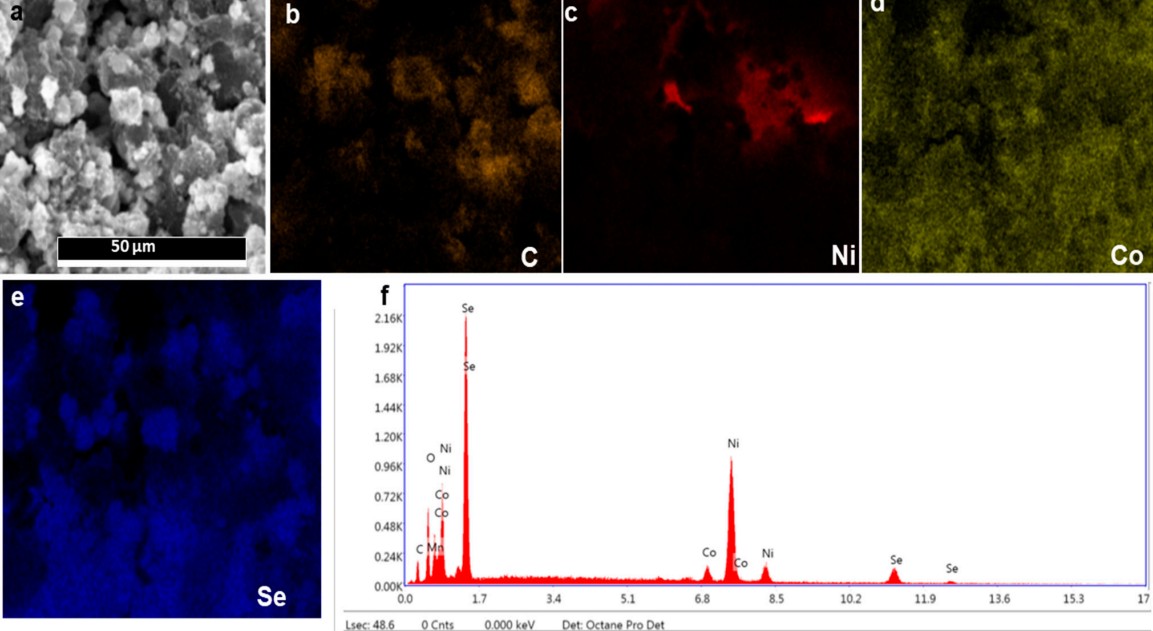

**Figure 3.** EDX analysis with elemental mapping for NiCoSe@CNTs with color scheme of different elements: (**a**) low magnified dark SEM, (**b**) carbon, (**c**) nickel, (**d**) cobalt, (**e**) selenium, and (**f**) relative abundance spectra.

To ascertain the elemental composition of materials, EDX (energy-dispersive X-ray spectroscopy) is a potent analytical method that is usually combined with scanning electron microscopy. EDX is a crucial tool for comprehending the distribution, concentration, and kind of elements present in composites, which are materials constructed from two or more constituent materials having noticeably differing physical or chemical properties. In Figure 3, the energy-dispersive X-ray (EDX) spectrum clearly exhibits distinct signals for Co, Ni, and Se elements. Moreover, the EDX spectrum also reveals the uniform embedding of these multi-metal elements in carbon nanotubes, indicating the successful preparation of the NiCoSe@CNTs composite, as shown in Figure 3. Nickel and carbon have the highest abundance, which represents that the composites have CNTs as the matrix and metal selenides as reinforcements within the synthesized materials. The respective elements are also represented by the mapping images with different colors, as shown in Figure 3a–e.

Figure 4 represents the manganese-based composite (MnCoSe@CNTs) with lower (Figure 4a) and higher (Figure 4a–c) resolution. Transition metal-based selenide nanoparticles are firmly anchored on the CNTs, resulting in the formation of the MnCoSe@CNTs composite. Upon the addition of Mn, the outer layer of the skeleton becomes smoother, potentially indicating the formation of a cube-like product at the surface.

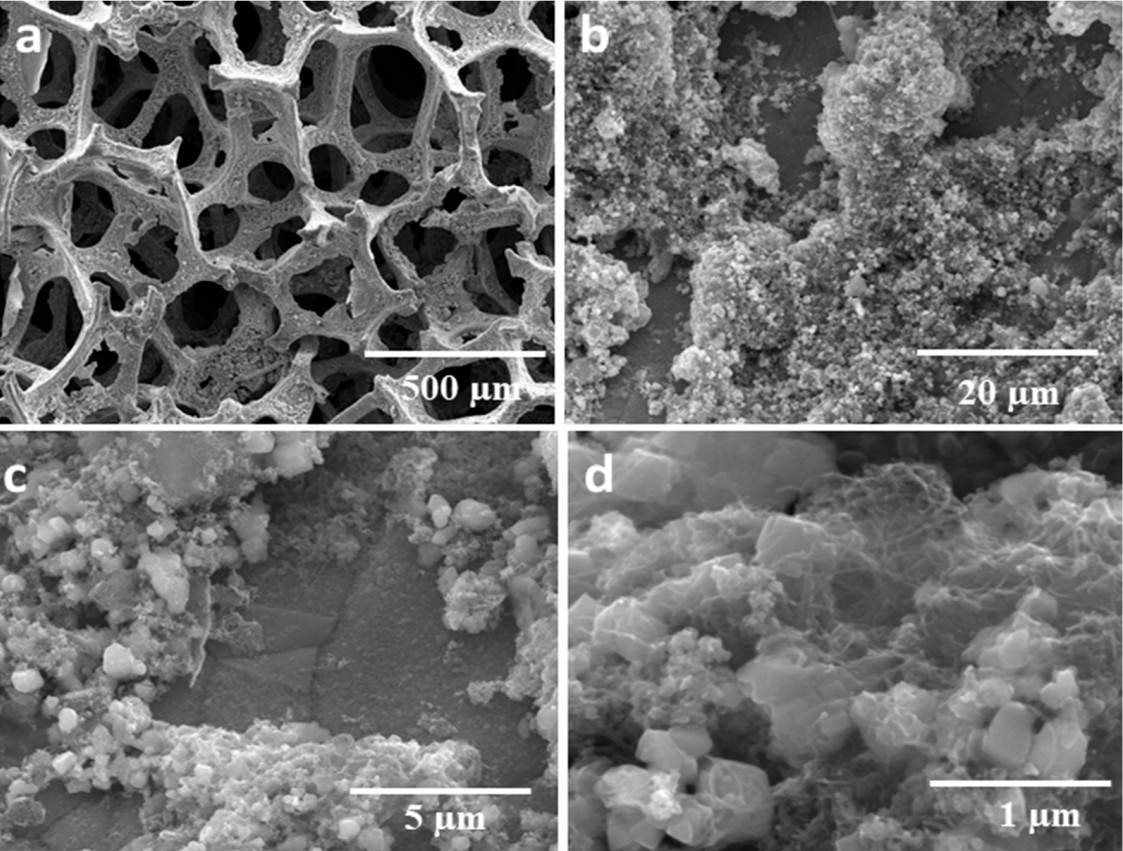

**Figure 4.** SEM mages for MnCoSe@CNTs modified NF with (**a**,**b**) lower and (**c**,**d**) higher magnifications.

Selenide, manganese, and cobalt have the highest abundance, which represents the composites that have CNTs as the matrix and metal selenides as reinforcements within the synthesized materials. The presence of oxygen might be from the substrate as we have deposited the composite materials and heated them in the oven, as shown in Figure 5.

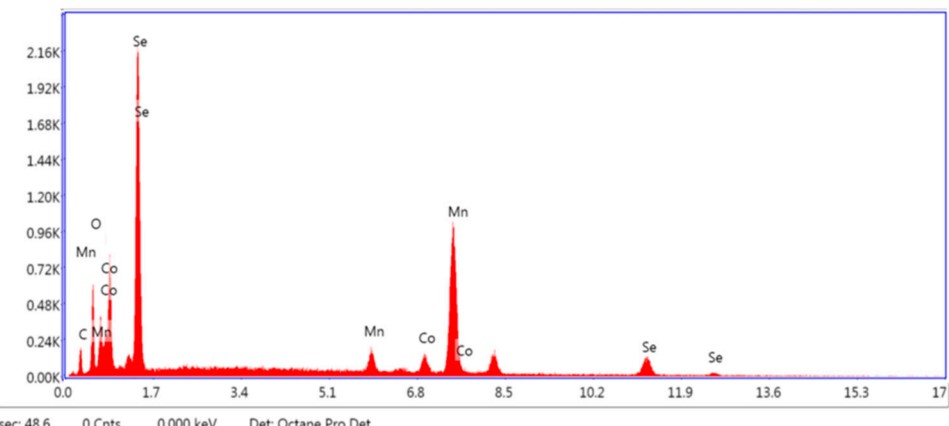

**Figure 5.** EDX analysis of MnCoSe@CNTs with the relative abundance of elements present in the composite.

### 3.2. Electrochemical Studies for OER

Two universally acknowledged kinetic parameters were selectively adopted to evaluate the electrocatalytic performance of the oxygen evolution reaction (OER): (i) onset overpotential (onset η), which marks the initiation of the reaction, and (ii) the overpotential needed to attain a current density of 10 mA/cm$^2$, indicating the efficiency at a specific current level [12]. In this study, the OER activity of fabricated electrodes, including NiCoSe/CNTs@NF, MnCoSe/CNTs@NF, CNTs@NF, and NF, was investigated in an alkaline solution. Figure 6a presents polarization curves of bare Ni foam and three free-standing Ni foam composites, and current density vs. applied potential to facilitate a comparative analysis of their respective OER activities. The MnCoSe/CNTs@NF composite exhibits a significant maximum current density with the lowest onset potential (1.57 V), demonstrating remarkable OER activity. In contrast, the NiCoSe/CNTs@NF composite achieved a much higher anode current density of 100 mAcm$^{-2}$ ($\eta_{100}$) with a remarkably low overpotential of 0.560 V (vs. RHE), distinctly lower than those of MnCoSe/CNTs@NF, blank NF, and CNTs@NF, which have corresponding values calculated as 0.612 V, 0.619 V, and 0.698 V, respectively. Additionally, the CNTs@NF electrode exhibits the smallest anodic current above 1.57 V and significantly lower OER activity, suggesting that the incorporation of CNTs into the nickel foam has minimal impact on the electrochemical reaction. A more detailed comparison of OER performances of selenide-based composites is provided in Table 1. Remarkably, the multilevel hexagonal hollow nanostructure referred to as (NiCoSe/CNTs@NF) exhibits a highly promising prospect for the electrocatalytic process, offering an abundance of accessible catalytic centers and synergistic effects from well-dispersed multimetallic composites, resulting in the maximized utilization of the electrocatalyst.

**Table 1.** Comparisons of the OER parameters of multimetallic selenides-based composites in strong alkaline solution.

| Electrocatalyst | $\eta_{100}$ (V) | Tafel Slop (mV/dec) | $C_{dl}$ (mFcm$^{-2}$) | $R_{ct}$ (Ω) |
|---|---|---|---|---|
| Ni foam | 0.619 | 312 | 2.52 | 2.66 |
| CNTs@ Ni foam | 0.698 | 282 | 1.09 | 1.50 |
| MnCoSe/CNTs@Ni foam | 0.612 | 177 | 7.07 | 1.03 |
| NiCoSe/CNTs@Ni foam | 0.560 | 163 | 7.24 | 0.90 |

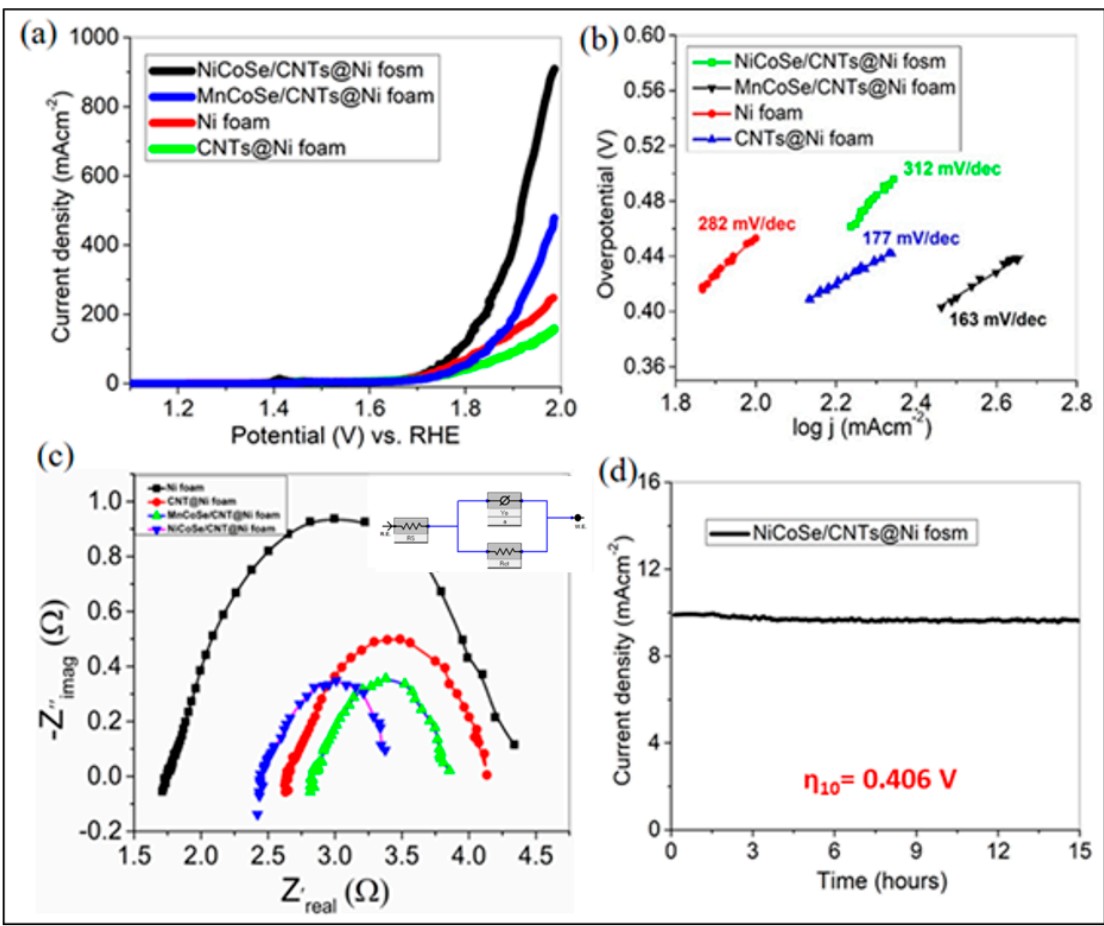

**Figure 6.** (**a**) Polarization curves recorded with *iR*-correction under alkaline conditions with a fixed scan rate of 5 mVs$^{-1}$. (**b**) Tafel plots of bare-NF and modified-NF with as-fabricated composites derived from OER voltammogram. (**c**) Nyquist plots of OER with frequencies ranging from 0.1 Hz to 1 MHz. (**d**) Chronoamperometric curve of NiCoSe/CNTs@NF with the applied overpotential of 0.406 V.

The Tafel slope of NiCoSe/CNTs@NF is approximately 163 mV/dec, which is lesser than MnCoSe/CNTs@NF (177 mVdec$^{-1}$), CNTs@NF (282 mVdec$^{-1}$), and blank NF (312 mVdec$^{-1}$) in (shown in Figure 6b), respectively. The lower Tafel slope value observed for NiCoSe/CNTs@NF confirms accelerated catalytic reaction kinetics for the charge transfer of OER on its surface. Figure 6c illustrates the Nyquist plots of EIS obtained from modeling via an equivalent circuit consisting of a series resistance ($R_s$) and charge transfer resistance ($R_{ct}$) in the frequency range from 105 Hz to 0.01 Hz. The Rct value is directly linked to the OER performance, as it represents the charge transfer impedance at the catalyst/ionic conducting interface. This information can be extracted from the low-frequency range of semicircles observed in Nyquist plots in Table 1, The NiCoSe/CNTs@NF composite exhibits a notably smaller $R_{ct}$ value (0.90 Ω) compared to MnCoSe/CNTs@NF (1.03 Ω), CNTs@NF (1.50 Ω), and blank NF (2.66 Ω). The smallest R$_{ct}$ value of MnCoSe/CNTs@NF signifies its distinguished electric conductivity and faster electron transport kinetics.

The electrochemically active surface area (ECSA) test was conducted under scan rates (50–600 mV/s) to evaluate the electrochemical specific surface area by measuring the double-layer capacitances ($C_{dl}$) in Figure 7a–d, with the corresponding Cdl curves shown in the inset. The C$_{dl}$ value of NiCoSe/CNTs@NF (7.24 mFcm$^{-2}$) is higher than that of MnCoSe/CNTs@NF (7.07 mFcm$^{-2}$) and CNTs@NF (1.09 mFcm$^{-2}$). Therefore, the NiCoSe/CNTs@NF composites have a larger electrochemical-specific surface area due to their large pore diameter, enabling more accessible sites for OER [51]. The chronoamperometric response recorded during the long-term stability test for the NiCoSe/CNTs@NF electrode at an overpotential of 0.406 V vs. RHE is shown in Figure 6d. The obtained result

indicated that the current density remained stable and maintained 97% of the original at 10 mAcm$^{-2}$. Furthermore, there was only a negligible decay in stability at low current density, demonstrating its superior durability.

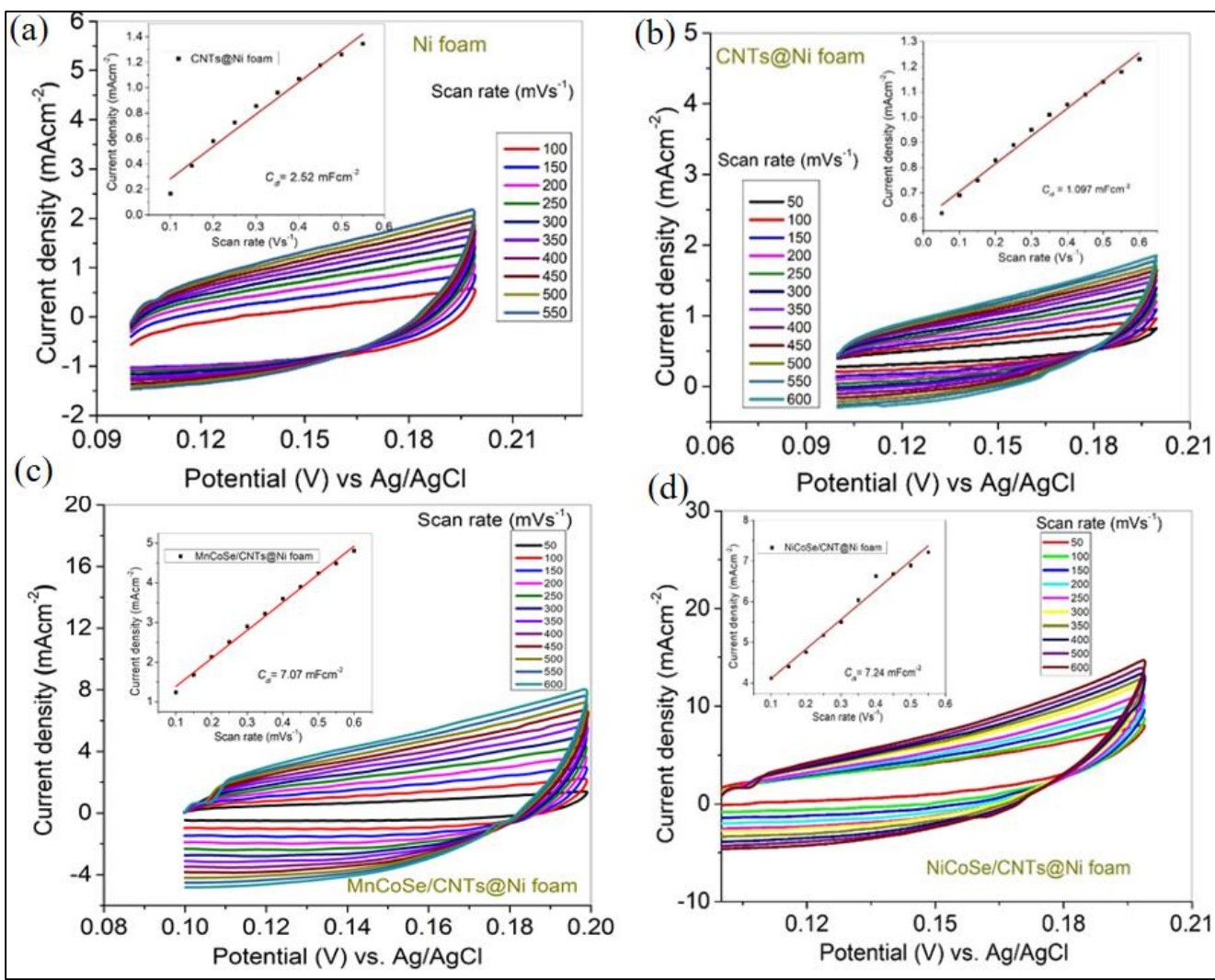

**Figure 7.** CV curves of (**a**) NF-bare, (**b**) CNT@NF, (**c**) MnCoSe/CNT@NF, and (**d**) NiCoSe/CNT@NF at different scan rates in the non-Faradaic region with corresponding $C_{dl}$ plots of the current density versus scan rates as inset in each figure.

## 4. Conclusions

Our investigation focused on the synthesis and characterization of ternary metal-based composites, including CNTs@NF, MnCoSe/CNTs@NF, and NiCoSe/CNTs@NF, as potential electrocatalysts for oxygen evolution reaction (OER) in alkaline electrolytes. The electrochemical measurements demonstrated that the NiCoSe/CNTs@NF composite exhibited remarkable OER activity with the lowest overpotential η (OER) of 0.560 V (vs. RHE), a Tafel slope of 163 mV/dec, and the smallest charge transfer impedance among all the investigated composites. This exceptional performance was attributed to the synergistic effects of well-dispersed multi-metal components and the enhanced charge transfer kinetics at the electrocatalyst/ionic conducting interface. Furthermore, the scanning electron microscopy (SEM) analysis provided valuable insights into the surface morphology and nanoarchitecture of the composites, confirming the successful formation of interconnected structures. Additionally, the energy-dispersive X-ray spectroscopy (EDX) analysis allowed us to verify the uniform distribution of the multi-metal components within the composites, further supporting their potential as efficient electrocatalysts. Further research on optimiz-

ing their composition and structure could lead to even more remarkable electrocatalytic performances and broader applications in the field of renewable energy.

**Author Contributions:** Conceptualization, M.S.A., S.R. and R.S.; methodology, Y.M., M.A., R.S. and A.A.; software, M.S.A. and R.S.; validation, A.A., R.S., S.A. and S.R.; formal analysis, A.A., N.A. and R.S.; investigation, M.S.A., S.R., R.S. and A.A.; resources, N.A. and A.A.; data curation, A.A. and S.R.; writing—review and editing. R.S., Y.M., M.I., N.H., S.R. and A.A.; writing—review and editing. R.S., Y.M., M.I., N.H., S.R. and A.A.; visualization, R.S., M.A., Y.M. and A.A.; supervision, A.A. and R.S.; project administration, N.A. and A.A.; funding acquisition, A.A. All authors have read and agreed to the published version of the manuscript.

**Funding:** This research was funded by Princess Nourah bint Abdulrahman University Researchers Supporting Project number (PNURSP2024R11), Princess Nourah bint Abdulrahman University, Riyadh, Saudi Arabia, and the Higher Education Commission (HEC) Pakistan for financial support under the startup research grant # 21-2146/SRGP/R&D/HEC2018.

**Institutional Review Board Statement:** Not applicable.

**Informed Consent Statement:** Not applicable.

**Data Availability Statement:** All data has been provided in the manuscript.

**Acknowledgments:** The authors express their gratitude to Princess Nourah bint Abdulrahman University Researchers Supporting Project number (PNURSP2024R11), Princess Nourah bint Abdulrahman University, Riyadh, Saudi Arabia. Authors are highly thankful to the Higher education commission (HEC) Pakistan and gratefully acknowledge the Università degli Studi di Cagliari. R.S. performed his activity in the framework of the International Ph.D. in Innovation Sciences and Technologies at the University of Cagliari, Italy. Elemental mapping was performed with support of PNA university.

**Conflicts of Interest:** The authors declare no conflicts of interest.

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
