# Peer review of "Carbon Nanotube Composites with Bimetallic Transition Metal Selenides as Efficient Electrocatalysts for Oxygen Evolution Reaction"

_sustainability, doi:10.3390/su16051953_

Round 1

Reviewer 1 Report

Comments and Suggestions for Authors

After careful review, I found that the manuscript has the following major issues that make it unsuitable for publication in this journal:

1.       Unit format errors: The manuscript contains several unit format errors, which not only affect the accuracy of the article but also reflect the author’s negligence in preparing the manuscript.

2.       Incorrect figures: The figures in the manuscript are not standardized, including but not limited to blurry images and unclear descriptions, which seriously affect the reader’s understanding of the content.

3.       Incorrect material performance description: The materials prepared in this paper lack innovation, and there are errors in the description of material properties. For example, when the current density reaches 100 mA cm-2, the overpotential is mistakenly described as 0.560 mV, while the correct value should be 0.560 V. Such errors not only lower the technical accuracy of the article but also affect the reader’s evaluation of the research results.

4.       Inconsistent overpotential description: The overpotential described in the abstract section is inconsistent with that described in the main text, causing readers to question the reliability of the article.

In conclusion, the current state of the manuscript does not meet the publication requirements of this journal, and it is recommended to reject the manuscript. It is hoped that the author will revise and resubmit or consider deepening the research before submitting again.

Comments on the Quality of English Language

Extensive editing of English language required.

Author Response

The authors are grateful to the Reviewer for the time devoted to the review of the manuscript as well as for the favourable opinion regarding the work. Considerations and suggestions provided in the report are extremely beneficial for clarifying a few fundamental issues.

Thank you again for your opinion about our work. All revisions have been made for improving quality of the paper by following your suggestions.

Reviewer 2 Report

Comments and Suggestions for Authors

The manuscript entitled "Carbon Nanotubes Composites with Bimetallic Transition Metal Selenides as Efficient Electrocatalysts for Oxygen Evolution Reactions" is interesting. I have some suggestions for improvement:

Abstract: there are other publications dealing with carbon nanotube-grafted core-shell-structured selenides and would be useful to beter highlith the novelty of present research

Introduction:

-line 62: "Recently, an engineer achieved a cost-effective oxygen evolution catalyst using earth-abundant transition elements" - could be reformulated

Experimental:

-line 129 - Scheme 1 is mentioned but it is not present in the manuscript

-line 133 - "4.0 mmol (or 28.55 mg, 0.12 mmol) of NiCl2.6H2O and CoCl2.6H2O (28.55 mg, 0.12 mmol)" - please correct

-line 176 -  please specify electrochemical impedance spectroscopy (EIS) registered with an amplitude of ...

Results and Discussions

- line 202 - elemental maping is mentioned but there are no images of elemental maping

Figure 2 - the scale is not very visible; please improve the image

-line 251 - "inset plots of current density vs applied potential to facilitate a comparative analysis" - but figure 6a does not have an inset

-line 252-254 - The NiCoSe/CNTs@NF composite ......  In contrast, the NiCoSe/CNTs@NF composite - please correct

At 3.2. Electrochemical studies for OER, some comparations with values obtained in other similar studies in literature could be given

Author Response

(The authors gave the same response as above.)

Reviewer 3 Report

Comments and Suggestions for Authors

In this manuscript, the authors reported the synthesis of bimetallic transition metal selenides compositing with carbon nanotubes and their application as efficient electrocatalysts for oxygen evolution reaction (OER) in alkaline solutions. Two types of bimetallic selenides were studied for this purpose and the NiCo based one showed better OER performance. These composites have the potential for use in large current density water splitting. The results can have implications for the further development of the water electrolysis technology for hydrogen generation. Overall, this work is worthy to be published in the journal Sustainability. However, some certain revision is required to further improve the manuscript. Please find below the detailed comments.

1. The authors concluded that “energy-dispersive X-ray spectroscopy (EDX) analysis allowed us to verify the uniform distribution of the multi-metal components within the composites”. However, currently only EDX spectra were provided. The mapping images should also be provided to support the claim that the elements were uniformly distributed within the composite.

2. The formation of composites is indeed an important strategy for developing OER catalysts. Further discussion related to this can be included in the Introduction and related works can be referenced (e.g., Small, 2021, 17, 2101573).

3. The authors stated that the overpotential to reach a current density of 100 mA/cm² (η100) is “0.560 mV (vs RHE)”. Please note that this statement was not scientifically sound. First of all, there is no need to include “vs RHE” since overpotential is defined as the difference between applied potential and equilibrium potential (hence unit should be either V or mV). Secondly, if overpotential is in the unit of V then the number should be 0.560, and if it is in the unit of mV then the number should be 560. Please revise the manuscript (Abstract, main text, Conclusion, and Table 1) according to the above comments.

4. To appeal to a broader readership, recent works on OER can be included in the Introduction (e.g., Small Methods, 2022, 6, 2201099; InfoMat 2023, DOI: 10.1002/inf2.12494).

5. Line 289-290, the Cdl values in this part of discussion were not consistent with those displayed in Figure 7 or Table 1. Firstly, the unit should be mF cm-2 and not uF cm-2. Secondly, the significant number for the Cdl value of CNTs@NF was not consistent in the figure (1.097) and main text (1.09).

6. For the stability test, the authors claimed that the potential could be maintained at a constant current density of 10 mA/cm2 at the potential of 1.5 V vs. RHE. However, in Figure 6d, the authors wrote an overpotential of 0.406 V for this stability. Please note that the potential value does not correspond to the overpotential value (given that equilibrium potential is 1.23 V).

7. Please note that the scale bars in Figure 2 were not clearly presented. Please revise to improve the visibility. The same applies for Figure 4.

8. Please note that for the EDX images in Figure 3 and Figure 5, the x axis and y axis should be provided.

9. The equivalent circuit in the inset of Figure 6c needs to be improved for better visibility.

Comments on the Quality of English Language

1. Some sentences need to be revised to improve their clarity. For instance, this sentence in Abstract appears to be quite confusing. “The synthesis process involved the facile one step hydrothermal growth of transition metal nanoparticles over the CNTs and acted as efficient electrode towards electrochemical water splitting.” Please check the structure of this sentence and improve the writing.

2. Some typos need to be corrected, such as “i(LSV)” in Line 166 and “i0.1 V to i0.20” in Line 173.

3. In the Title and Abstract, “oxygen evolution reactions” should be revised into “oxygen evolution reaction”.

Author Response

(The authors gave the same response as above.)

Round 2

Reviewer 1 Report

Comments and Suggestions for Authors

The manuscript has been comprehensively upgraded. The author can refer to the relevant literature, which will be beneficial to the quality of the article. 10.1016/j.jmst.2023.02.050; 10.1016/j.ceramint.2022.09.035; 10.1016/j.nanoen.2022.107876

Comments on the Quality of English Language

Minor editing of English language required.

Author Response

Thank you again for your opinion about our work. All revisions have been made for improving quality of the paper by following your suggestions.

Reviewer 2 Report

Comments and Suggestions for Authors

The manuscript has been improved and can be accepted for publication.

Author Response

The authors are thankful to reviewer for their opinion to Academic editor for manuscript acceptance for publication.